⊕ | **Open Peer Review** | Immunology | Research Article

# Short- and long-term stability of SARS-CoV-2 antibodies on dried blood spots under different storage conditions

Eline Meyers,[1] Anja Coen,[2] Elizaveta Padalko,[1,3] Piet Cools[1]

**ABSTRACT** Dried blood spots (DBS) are broadly used for different serological analyses, including severe acute respiratory syndrome coronavirus 2 (SARS-CoV-2) antibody assessment. In order to biobank DBS samples, an understanding of the preservation of SARS-CoV-2 antibodies is needed. Therefore, we assessed the stability of SARS-CoV-2 antibodies on DBS during short- and long-term storage under different storage temperatures. Seven sample donors were enrolled, each donating twenty 6 mm DBS to assess anti-spike (S1) SARS-CoV-2 IgG antibodies (EUROIMMUN). Baseline samples were analyzed on the day of collection. The remainder of the samples was stored in grip seal bags kept in a cryobox at room temperature/4°C until 2 months after collection and at −20°C until 2 years after collection. Samples were analyzed at regular intervals within the total storage duration and after one and five freeze–thaw cycles. A pooled coefficient of variation was calculated for each storage temperature. We found that anti-S1 SARS-CoV-2 antibodies collected on DBS saver cards remain stable during short-term storage at RT, 4°C, and −20°C (at least to 2 months) and long-term storage at −20°C (at least 2 years). Moreover, up to five freeze–thaw cycles can occur without impacting the anti-S1 SARS-CoV-2 antibody level. The inter-assay coefficient of variation lies between 10 and 15%. As DBS can be preserved for both shorter periods of time at RT and longer periods of time at −20°C, they are a perfect application for studies that require sample shipment by mail, self-sampling studies, studies in limited resource settings and biobanking.

**IMPORTANCE** Dried blood spots (DBS) are currently widely used as a microsampling technique for different qualitative and quantitative serological assessments. Yet, there is a lack of long-term stability and storage condition studies. In our study, first, we assessed the stability of SARS-CoV-2 antibodies on DBS up to 2 years post-collection. We believe that our data are not only important for future COVID-19 research but also for studies on other infections/diseases using DBS-based serology.

**KEYWORDS** SARS-CoV-2 antibodies, dried blood spots, storage, preservation, freeze-thaw

D ried blood spots (DBS) are broadly used for different serological analyses, as they require minimally invasive blood sampling (collected through a fingerprick) and are inexpensive and easy to process (1–5). Also, during the coronavirus disease (COVID-19) pandemic, DBS sampling was shown to be valid for qualitative and (semi-)quantitative assessments of SARS-CoV-2 antibodies (6–10). Multiple studies in different countries have (longitudinally) collected DBS in large study populations (11–16). Within these studies, samples are often shipped between sites and/or stored for a shorter or longer period of time between collection and analysis. However, for these applications, it is important to understand the stability of SARS-CoV-2 antibodies in DBS over time and the optimal storage temperature. Previous studies have already shown that antibodies directed against other viruses (e.g., measles, varicella zoster, and rubella) remain stable at

Address correspondence to Piet Cools, piet.cools@ugent.be.

The authors declare no conflict of interest.

See the funding table on p. 7.

4°C up to at least 4–6 months (17, 18) and at least up to 1 month at RT (19). Moreover, it has been shown that anti-HIV-1 antibodies can be preserved at −20°C for several years without significant loss (20). Similarly, recent studies have been investigating the stability of SARS-CoV-2 antibodies on DBS for short-term storage under different conditions (21–25). However, data on long-term storage of DBS for the detection and quantification of SARS-CoV-2 antibodies are lacking. Therefore, in the present study, we aimed to assess the stability of SARS-CoV-2 antibodies in DBS for short-term storage (up to 2 months) at RT, 4°C, and −20°C and long-term storage (up to 2 years) at −20°C. Additionally, we aimed to investigate the effect of up to five freeze–thaw cycles on SARS-CoV-2 antibody stability.

## MATERIALS AND METHODS

### Study subjects and sampling procedures

In July 2021, sample donors were enrolled at Ghent University ($n = 7$). Donor capillary blood was collected on DBS saver cards (EUROIMMUN, Lübeck, Germany) until 20 spots were filled (sufficient to punch twenty 6 mm circles) by means of one to three finger-pricks (SARSTEDT AG & Co., Nümbrecht, Germany). Six out of seven sample donors were COVID-19 vaccinated, and therefore, expected to be SARS-CoV-2 seropositive.

### Storage conditions

Baseline DBS were analyzed on the day of collection. The remainder of the DBS card was stored in grip seal bags (Staples, Grimbergen, Belgium) kept in a cardboard cryobox under the respective storage condition [room temperature (RT, 18–25°C), 4°C (2–4°C), or −20°C (−18−−22°C)]. No desiccant was used. There was no control for humidity levels under any storage condition. Also, the temperature for storage at RT was not controlled but was assumed to be between 18 and 25°C indoor during Western European weather conditions. Samples stored at RT and 4°C were analyzed for 14 days, 1 month, and 2 months post-collection. Samples stored at −20°C were analyzed at 14 days monthly up to 6, 10, 12, 14, 18, and 24 months post-collection. To investigate the effect of one and five freeze–thaw cycle(s), samples were frozen first at −20°C for at least 6 h and thereafter thawed for 1 h at room temperature (repeated five times for five freeze–thaw cycles). Samples that underwent freeze–thaw cycles were analyzed within 1 week after collection.

### Detection of SARS-CoV-2 antibodies

DBS were analyzed for the presence of anti-spike (S1 antigen) SARS-CoV-2 IgG antibodies using ELISA (EUROIMMUN; PerkinElmer Health Sciences, Inc.), as previously described and validated (6). One circle (6 mm diameter) was cut out from each DBS card using a puncher and was placed in a well of a sterile 96-well U-shaped microtiter plate. A total volume of 250 µL preheated (1 h at 37°C) sample buffer was added to each sample well of the 96-well microtiter plate, and the plate was incubated at 37°C for 1 h. After gentle mixing of the eluate by means of up-and-down pipetting, a total of 100 µL of this eluate was used for ELISA manually performed according to the manufacturer's instructions. The ELISA procedures were conducted by different operators over time. Optical density was measured at 450 nm (reference wavelength 650 nm) on the Behring ELISA Processor III (Siemens AG, Munich, Germany). Each run included an internal positive control (human IgG) and a negative control. For the analysis at 18 and 24 months, OD was measured at 450 nm (reference wavelength 650 nm) on the Magellan Sunrise absorbance microplate reader (Tecan Group Ltd., Männedorf, Switzerland) after validation of the new reader (see supplemental file). Anti-S1 SARS-CoV-2 IgG OD ratios were calculated by dividing the raw OD value by the mean value of the raw OD value of the calibrators (run in duplicate, reference OD of 0.365, valid when >0.140) in Microsoft Excel (Microsoft, Redmond, WA, USA). DBS were classified according to their anti-S1

SARS-CoV-2 IgG OD ratio (450 nm) as negative (<0.8), borderline (0.8–1.0), or positive (≥1.1), as recommended by the manufacturer.

## Statistical analysis

To assess the effect of long-term storage on anti-S1 SARS-CoV-2 IgG OD ratio per storage condition, we applied a mixed-effects model with storage time as a fixed effect and subject as a random effect. To assess the effect of both (short-term) storage and storage condition (RT, 4°C, −20°C) on the anti-S1 SARS-CoV-2 IgG OD ratio, we applied a mixed-effects model, with storage time, storage condition, and storage time × storage condition as fixed effects and subject as random effect. Bonferroni correction was applied for multiple comparisons. Residuals were normally distributed as checked by QQ plots. To assess inter-assay variability between the different repeated measures over time, we calculated the coefficient of variation (CV) per sample and per storage condition. First, we divided the standard deviation by the mean value of the repeated measures per sample and storage condition ($n = 4$ for RT/4°C; $n = 12$ for −20°C). Second, for each storage condition, we calculated pooled CVs by taking the mean of the CV obtained for each sample. $P$-values < 0.05 were considered statistically significant.

All statistical analyses were performed in GraphPad Prism 9.3.1 (GraphPad Software, San Diego, California, USA).

## RESULTS

### Short-term stability of anti-S1 SARS-CoV-2 IgG on DBS at room temperature, 4°C, and −20°C

We determined the anti-S1 SARS-CoV-2 IgG OD ratio in DBS at baseline and after storing them for 14 days, 1 month, and 2 months under different conditions (RT, 4°C, and −20°C). All samples were positive for anti-S1 SARS-CoV-2 IgG, except for one, which was the sample from the non-vaccinated donor. The anti-S1 SARS-CoV-2 IgG OD ratio did not differ significantly from baseline at the respective timepoints under any condition ($P$ > 0.05), see Fig. 1A to C. Moreover, no differences between storage conditions were observed (Fig. 1D) ($P$ > 0.05). Similar observations were made for the stability of antibody concentrations expressed in international units/mL (see supplemental file).

### Long-term stability of anti-S1 SARS-CoV-2 IgG in DBS at −20°C

To assess the long-term stability of anti-S1 SARS-CoV-2 IgG in DBS, we stored DBS at −20°C and assessed the anti-S1 SARS-CoV-2 IgG OD ratio at baseline and different timepoints up until 24 months after collection. No statistically significant differences were observed in the anti-S1 SARS-CoV-2 IgG OD ratio between baseline and the respective timepoints, except for month 4 ($P$-value = 0.02) (Fig. 2). Similar observations were made for the stability of antibody concentrations expressed in international units/mL (see supplemental file).

### Effect of one and five freeze–thaw cycles on the stability of anti-S1 SARS-CoV-2 IgG on DBS

We investigated the effect of one and five freeze–thaw cycles on the stability of anti-S1 SARS-CoV-2 IgG in DBS (Fig. 3). No statistically significant differences were found in the mean anti-S1 SARS-CoV-2 IgG OD ratio after one or five freeze–thaw cycles compared to baseline ($P$ > 0.05).

### Inter-assay variability of detection of anti-S1 SARS-CoV-2 antibodies on DBS

We determined the CV for anti-S1 SARS-CoV-2 antibody detection on DBS stored under different conditions as a measure for inter-assay variability (Table 1). Pooled CVs range between 10 and 15% for all three conditions (RT/4°C/−20°C).

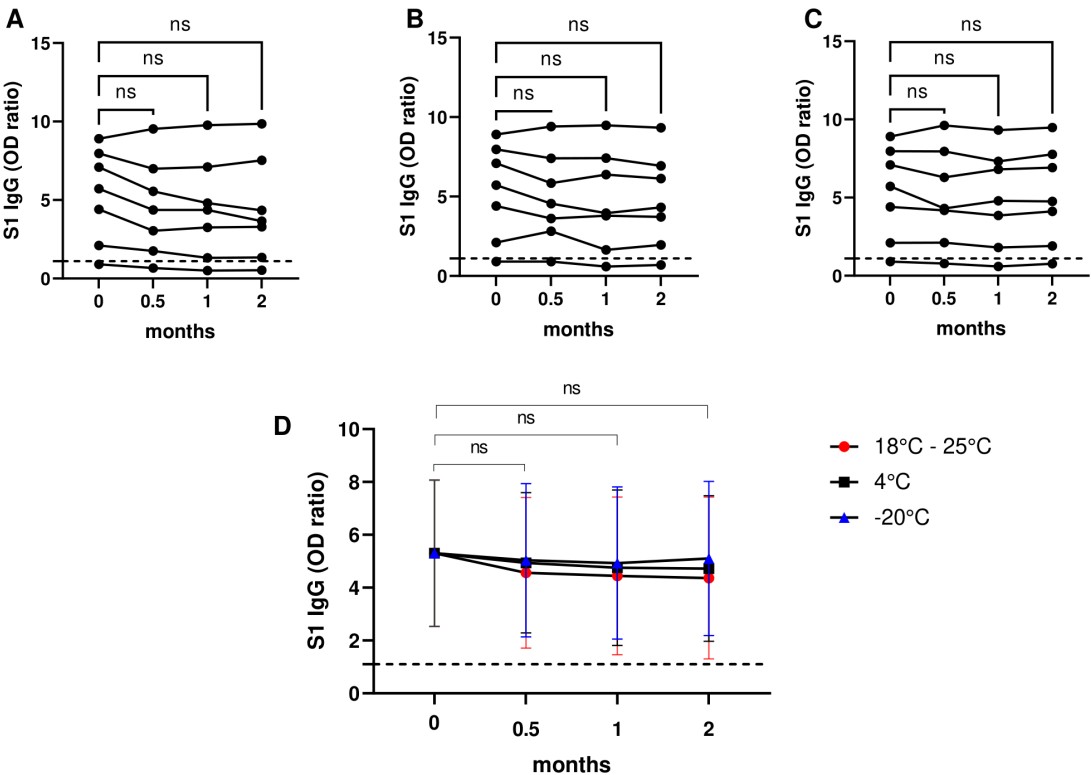

**FIG 1** Short-term stability of anti-S1 SARS-CoV-2 IgG in dried blood spots (DBS) ($n = 7$). Optical density (OD) values are shown for DBS at baseline and stored 14 days, 1 month, and 2 months after collection at room temperature (A), 4°C (B), and −20°C (C). (D) represents the mean S1 IgG OD ratio with 95% confidence intervals (error bars) of all donor DBS per storage condition (shown by red circle, black square, and blue triangle) over time. ns: non-significant, $P > 0.05$. The ticked horizontal line represents the cut-off for SARS-CoV-2 seropositivity.

## DISCUSSION

We assessed the stability of anti-S1 SARS-CoV-2 antibodies in DBS during short-term storage at RT, 4°C, and −20°C and long-term storage at −20°C. Our findings show that SARS-CoV-2 antibodies collected on DBS saver cards remain stable under all investigated conditions (at RT and 4°C for at least 2 months and at −20°C for at least 2 years). Moreover, we showed that up to five freeze–thaw cycles can occur without impacting the anti-S1 SARS-CoV-2 IgG OD ratio, being an important aspect as frozen DBS samples tend to thaw very rapidly during handling at RT. Additionally, we showed that the inter-assay CV lies between 10 and 15%, which is considered as normal for tests performed over a long period of time by different operators (26, 27).

Although we found one-time point (after 4 months of storage at −20°C) to be statistically significantly different from baseline ($P = 0.02$), the timepoints thereafter were not, meaning that the lower antibody ratios were probably due to inter-assay variation instead of due to preservation loss. Moreover, by interpolation of the anti-S1 SARS-CoV-2 IgG OD ratios to antibody concentrations in international units/mL, we demonstrated that no clinically relevant changes in antibody levels occur (e.g., falling below the cut-off for seropositivity) when stored on DBS for a longer period.

Other studies have similarly reported that SARS-CoV-2 antibodies remain stable in DBS during short-term storage (ranging from 28 up to 200 days) when stored at 4°C, −20°C, and/or RT (21–24). Nevertheless, we are the first to study the stability of SARS-CoV-2 antibodies on DBS for up to 2 years. In our study, we found that all investigated temperatures (RT, 4°C, −20°C) provide equal preservation of SARS-CoV-2 antibodies on

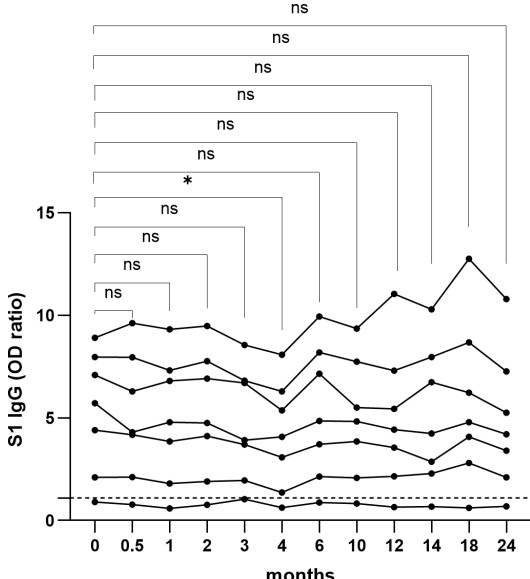

**FIG 2** Stability of anti-S1 SARS-CoV-2 IgG on dried blood spots ($n = 7$) stored for up to 24 months at −20°C. ns: non-significant, $P > 0.05$.; *: $P = 0.02$. The ticked horizontal line represents the cut-off for SARS-CoV-2 seropositivity.

DBS during short-term storage. This simplifies sample handling and logistics, that is, in contrast to venous blood, samples can be transported and processed at RT.

Although this study provides important information for experiments using DBS for SARS-CoV-2 detection, which have been preserved for longer periods of time, it is important to note that this study is limited by its small sample size ($n = 7$). Additionally, we have not investigated the effect of high temperatures (>25°C) on SARS-CoV-2 stability on DBS. This should be taken into account when considering the use of DBS for studies assessing SARS-CoV-2 antibodies that require mail shipment. Nevertheless, others demonstrated that SARS-CoV-2 antibodies on DBS are also stable for short-term storage at 37–40°C (22–24). Only when DBS are stored at temperatures ≥ 55°C or temperatures ≥ 29°C combined with highly humid conditions (~99%), a significant loss in SARS-CoV-2 antibodies can be observed (22–24).

Altogether, our data demonstrate the reliability and robustness of DBS saver cards for the preservation of SARS-CoV-2 antibodies. As DBS can be stored at RT for at least 2 months, they are a perfect application for studies that require sample shipment by mail, self-sampling studies, and studies in limited resource settings. Moreover, as they can be

**TABLE 1** Overview of the coefficients of variation (CV) for anti-S1 SARS-CoV-2 detection in dried blood spots stored under different conditions (room temperature, 4°C, and −20°C) as a measure for inter-assay variability OD: optical density

|  | Baseline OD ratio | CV RT[a] (%) | CV 4°C[a] (%) | CV −20°C[b] (%) |
|---|---|---|---|---|
| Sample 1 | 5.72 | 16.46 | 14.20 | 10.16 |
| Sample 2 | 2.1 | 19.83 | 20.33 | 15.32 |
| Sample 3 | 8.9 | 3.94 | 2.42 | 12.23 |
| Sample 4 | 7.97 | 5.21 | 4.95 | 8.12 |
| Sample 5 | 4.41 | 15.23 | 7.93 | 11.67 |
| Sample 6 | 0.9 | 24.30 | 17.28 | 17.04 |
| Sample 7 | 7.1 | 19.13 | 7.29 | 10.97 |
| Pooled CV |  | 14.87 | 10.63 | 12.22 |

[a]Based on four independent measurements.
[b]Based on 12 independent measurements.

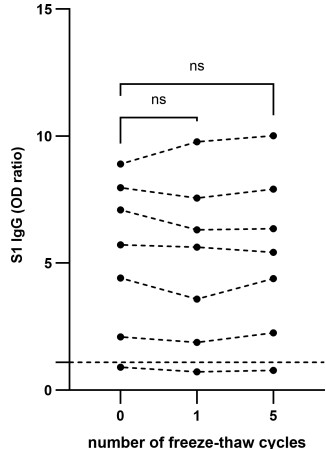

**FIG 3** Effect of one and five freeze–thaw cycles on the stability of anti-S1 SARS-CoV-2 IgG in dried blood spots ($n = 7$). ns: non-significant, $P > 0.05$. The ticked horizontal line represents the cut-off for SARS-CoV-2 seropositivity.

preserved for longer periods of time at −20°C, they can be of great added value in studies that make use of long-term preserved samples.

## Conclusion

DBS are broadly applied for the qualitative and quantitative assessments of SARS-CoV-2 antibodies; however, it is important to understand the stability of SARS-CoV-2 antibodies in DBS over time and the optimal storage temperature.

Our research shows that both short-term storage of DBS at room temperature, 4°C, and −20°C for at least up to 2 months and long-term storage at −20°C for at least up to 2 years maintain SARS-CoV-2 antibody stability. Moreover, up to five freeze–thaw cycles can occur without impacting the SARS-CoV-2 antibody level. These findings have significant implications, making DBS an ideal choice for sample shipment by mail, self-sampling studies, research in resource-limited settings, and long-term preserved samples. Therefore, this work contributes to the growing understanding of DBS' practical utility in serological analyses, especially in the context of SARS-CoV-2 antibody detection.

## ACKNOWLEDGMENTS

The authors wish to thank the sample donors for participating in this study.

This study was funded by the Special Research Fund of Ghent University (BOF.COV.2020.0010.01) and the Research Foundation Flanders (1SD2524N).

Conceptualization: P.C., E.M.; data curation: E.M.; formal analysis: E.M.; funding acquisition: P.C., E.M.; investigation: E.M.; A.C.; methodology: E.M.; A.C.; project administration: E.M.; validation: P.C.; visualization: E.M.; writing original draft: E.M.; and writing review and editing: P.C., E.P., A.C. All authors have read and agreed to the published version of the manuscript.

## AUTHOR AFFILIATIONS

[1]Department of Diagnostic Sciences, Faculty of Medicine and Health Sciences, Ghent University, Ghent, Belgium
[2]Department of Public Health and Primary Care, Faculty of Medicine and Health Sciences, Ghent University, Ghent, Belgium
[3]Laboratory of Medical Microbiology, Ghent University Hospital, Ghent, Belgium

## AUTHOR ORCIDs

Eline Meyers ⬤ http://orcid.org/0000-0002-3549-9548
Piet Cools ⬤ http://orcid.org/0000-0003-2980-5307

## FUNDING

| Funder | Grant(s) | Author(s) |
| --- | --- | --- |
| Special Research Fund Ghent University | BOF.COV.2020.0010.01 | Piet Cools |
| Research Foundation Flanders | 1SDN2524N | Eline Meyers |

## AUTHOR CONTRIBUTIONS

Anja Coen, Investigation, Methodology, Writing – review and editing | Elizaveta Padalko, Writing – review and editing | Piet Cools, Conceptualization, Funding acquisition, Validation, Writing – review and editing.

## DATA AVAILABILITY

The data are available in Tables S1 to S3 included in the supplemental files here: https://doi.org/10.5281/zenodo.13788826.

## ETHICS APPROVAL

The current study was approved by the Ethical Committee of the Ghent University Hospital (BC-07665) and conducted in accordance with the Declaration of Helsinki. Informed consent was obtained from every participant.

## ADDITIONAL FILES

The following material is available online.

### Open Peer Review

**PEER REVIEW HISTORY (review-history.pdf).** An accounting of the reviewer comments and feedback.

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
