## [Reviewer comments · Microbiology Spectrum]

Microbiology Spectrum

Short- and long-term stability of SARS-CoV-2 antibodies on Dried Blood Spots under different storage conditions

Eline Meyers, Anja Coen, Elizaveta Padalko, and Piet Cools

Corresponding Author(s): Eline Meyers, Universiteit Gent

Review Timeline:

Submission Date:	May 3, 2024
Editorial Decision:	August 23, 2024
Revision Received:	September 5, 2024
Accepted:	September 8, 2024

Editor: Yi-Chin Fan

Reviewer(s): The reviewers have opted to remain anonymous.

Transaction Report:

DOI: <https://doi.org/10.1128/spectrum.01113-24>

Re: Spectrum01113-24 (Short- and long-term stability of SARS-CoV-2 antibodies on Dried Blood Spots under different storage conditions)

Dear Dr. Eline Meyers:

Thank you for the privilege of reviewing your work. Below you will find my comments, instructions from the Spectrum editorial office, and the reviewer comments.

Revision Guidelines

Sincerely,
Yi-Chin Fan
Editor
Microbiology Spectrum

Reviewer #1 (Comments for the Author):

This study is important especially for biobanking. The authors assessed the stability of SARS-CoV-2 antibodies on DBS during short- and long-term storage under different storage temperatures. Anti-S1 SARS-CoV-2 antibodies collected on DBS saver cards remained stable during short-term storage at RT, 4{degree sign}C and -20{degree sign}C (at least to two months) and long-term storage at -20{degree sign}C (at least two years). Moreover, up to five freeze-thaw cycles can occur without impacting

the anti-S1 SARS-CoV-2 antibody level.

There are some comments and suggestion included in the attached pdf.

[revised manuscript text omitted]

368

369

- 18°C - 25°C
- 4°C
- ▲ -20°C

number of freeze-thaw cycles

Response to Reviewer

We thank the Reviewer for taking the time to review our manuscript and give suggestions to improve it. Please find below our point-by-point response.

Line 67-72: The aim of the study is better summarized to one or two aims only. The CV measurement is part of the analysis which should be done

We reduced the aim of the study to two sentences, leaving out the part about CV measurement.

Line 90: how RT and humidity were assessed?

We did not control for humidity at any storage condition, nor did we control for temperature for storage at room temperature. I clarified this in the manuscript.

Line 112: ELISA instead of Elisa

Adjusted.

Line 113: what is the internal positive control?

Human IgG, added to the manuscript.

Line 118: what is the concentration range of calibrators used?

The calibrator has a reference of OD of 0.365, and is considered valid when >0.140. This was added to the manuscript.

Line 131: write full name please

Adjusted.

Line 132: how many measurement per sample?

For RT and -4°C, 4 measurements per sample were done (short-term follow-up). For -20°C, 12 measurements per sample were done (long-term follow-up). This was added to the manuscript and is also mentioned in the legend of table 1.

Line 145: Do you already know the status of the donors using serum for example?

We did not assess the serostatus of the donors prior to sampling, however, six out of seven received a COVID-19 vaccine, and were therefore expected to be seropositive. This is mentioned in line 81.

Re: Spectrum01113-24R1 (Short- and long-term stability of SARS-CoV-2 antibodies on Dried Blood Spots under different storage conditions)

Dear Dr. Eline Meyers:

Your manuscript has been accepted, and I am forwarding it to the ASM production staff for publication. Your paper will first be checked to make sure all elements meet the technical requirements. ASM staff will contact you if anything needs to be revised before copyediting and production can begin. Otherwise, you will be notified when your proofs are ready to be viewed.

Sincerely,
Yi-Chin Fan
Editor
Microbiology Spectrum